# Acupuncture Improves Heart Rate Variability, Oxidative Stress Level, Exercise Tolerance, and Quality of Life in Tracheal Collapse Dogs

**DOI:** 10.3390/vetsci9020088

**Published:** 2022-02-18

**Authors:** Phurion Chueainta, Veerasak Punyapornwithaya, Weerapongse Tangjitjaroen, Wanpitak Pongkan, Chavalit Boonyapakorn

**Affiliations:** 1Veterinary Cardiopulmonary Clinic, Small Animal Hospital, Faculty of Veterinary Medicine, Chiang Mai University, Chiang Mai 50200, Thailand; c.phurion@gmail.com (P.C.); wanpitak.p@cmu.ac.th (W.P.); 2Department of Food Animal Clinic, Faculty of Veterinary Medicine, Chiang Mai University, Chiang Mai 50100, Thailand; veerasak.p@cmu.ac.th; 3Department of Companion Animal and Wildlife Clinic, Faculty of Veterinary Medicine, Chiang Mai University, Chiang Mai 50100, Thailand; weerapongse.t@cmu.ac.th; 4Department of Veterinary Biosciences and Veterinary Public Health, Faculty of Veterinary Medicine, Chiang Mai University, Chiang Mai 50100, Thailand; 5Integrative Research Center for Veterinary Preventive Medicine, Faculty of Veterinary Medicine, Chiang Mai University, Chiang Mai 50100, Thailand

**Keywords:** tracheal collapse, heart rate variability, acupuncture, dogs

## Abstract

Among the respiratory disorders in dogs from small breeds, tracheal collapse is one of the most commonly found in clinical practice. Presently, acupuncture is widely used as an alternative treatment which was shown to bring about positive effects in the treatment of human respiratory diseases. The present study demonstrated the effect of acupuncture on tracheal collapse dogs. We hypothesized that acupuncture can help dogs suffering from tracheal collapse by improving various parameters including heart rate variability, serum biomarkers for oxidative stress, exercise performance, and quality of life. Twenty client-owned dogs from small breeds with tracheal collapse disease were enrolled. The study was divided into two 5-week periods. During the first period, the dogs received normal veterinary care but received no acupuncture treatment (NAC). After completing that period, all forms of treatment were withheld for one week before the beginning of the second period. In the second period, all dogs restarted normal veterinary care and underwent acupuncture treatment (AC) once a week for five consecutive weeks. Blood was collected at the beginning and end of each of the two periods for malondialdehyde (MDA) level measurement. Heart rate variability (HRV) was recorded at the 1st, 3rd and 5th weeks of both periods. Exercise tests were performed at the beginning and end of AC period and questionnaire interviews with the owners were accomplished at the end of each period. The results showed that acupuncture can alleviate clinical signs of tracheal collapse, reduce MDA level, and improve sympathovagal balance. We suggest that acupuncture treatment could be used as an adjunct treatment for canine tracheal collapse.

## 1. Introduction

Tracheal collapse is one of the most common respiratory problems found in dogs from small breeds [1]. Softening of airway cartilage can affect both the upper and lower airway and can lead to the collapse of the airway. The definitive cause of tracheal collapse is unclear, but it has been suggested that it is related to congenital or secondary chronic inflammation [1,2,3,4]. The specific clinical signs depend on the severity of the collapse, including a paroxysmal goose-honk cough, various degrees of respiratory difficulty, and noisy breathing which is usually exacerbated by excitation [5]. A previous study found that 72% of dogs suffering from airway collapse showed evidence of inflammation in bronchoalveolar lavage fluid [1]. The inflammatory process causes an increased expression of oxidative stress and inflammatory biomarkers [6]. Excessive reactive oxygen species (ROS) resulting from oxidative stress can cause tissue damage by a variety of mechanisms including apoptosis, autophagy, and ferroptosis. These mechanisms finally lead to cellular degeneration and death [7].

Heart rate variability (HRV) refers to the variation in the beat-to-beat or R-R interval. HRV reflects the activity of the cardiorespiratory control system and can be used to detect variations of the sympathetic and parasympathetic nervous systems. The status of autonomic nervous system (ANS) balance can provide prognostic information for several diseases [8,9,10,11,12,13,14,15,16,17,18,19,20,21,22,23,24]. HRV impairment has been reported in humans with chronic obstructive pulmonary disease (COPD) [25,26,27]. HRV can also predict the prognosis and the mortality risk in patients with acute respiratory distress syndrome [28]. A previous study also demonstrated that people with an increased HRV are at a lower risk of developing an abnormal health condition such as cardiovascular diseases [29]. Moreover, impairment of HRV was reported to be related to an increase of ROS in the oxidative stress process, resulting in greater oxidative stress.

Currently, acupuncture (AC), an alternative medical treatment originating from traditional Chinese veterinary medicine (TCVM), has become widely accepted in veterinary practice [30]. Treatment involves stimulation of specific points on the patient’s body surface called acupoints [31]. There have been many studies in both human and veterinary medicine reporting the benefits and the positive therapeutic effects of AC [30,31,32,33,34,35,36,37,38,39,40,41]. Many studies of AC in respiratory disease treatment have found that it improves lung function in both humans and rats [42,43], reduces dyspnea, and improves the quality of life of the patient [44]. However, the effects of AC treatment on HRV parameters, oxidative stress, and exercise performance in tracheal collapse dogs have never been investigated. The aim of this study was to investigate the effects of acupuncture on all these aspects of the health state of the tested dogs. We hypothesized that acupuncture can improve parameters of HRV, serum biomarkers for oxidative stress, exercise performance, and quality of life in tracheal collapse dogs.

## 2. Materials and Methods

### 2.1. Animals and the Experiment Protocol

Twenty client-owned dogs from small breeds with tracheal collapse disease were enrolled at the Cardiopulmonary Clinic, Faculty of Veterinary Medicine, Chiang Mai University. There were Pomeranians (*n* = 11, 55%), Chihuahuas (*n* = 3, 15%), poodles (*n* = 2, 10%), Yorkshire terriers (*n* = 2, 10%), and pugs (*n* = 2, 10%). Body condition score (BCS) was determined using 9-scale system. Tracheal collapse was diagnosed based on history taking, clinical signs, physical examinations, and radiographic images. Dogs were excluded if they had any abnormalities compatible with systemic diseases, inflammation/infection, or neoplastic diseases based on the history of illness, physical examination, or abnormalities in hematology and/or blood chemistry profiles.

This 11-week study was divided into two 5-week periods separated by a one-week rest period. During the first period, no acupuncture treatment (NAC) was given. All 20 tracheal collapse dogs in both groups were given medical treatment: 0.625 mg terbutaline sulfate (President Inter Pharma Co., Ltd., Nonthaburi, Thailand) and 0.5 mg/kg of dextromethorphan HBr (Greater Pharma Manufacturing Co., Ltd., Bangkok, Thailand) orally, twice a day. After completing the NAC treatment period, medication for all dogs was withheld for one week before beginning the second (experimental) period. During that period, the acupuncture treatment period (AC), all dogs received the same medication as during the first period, but acupuncture treatment was added once a week for five consecutive weeks. Blood collection for malondialdehyde (MDA) level measurement was carried out at the beginning and end of each period. HRV was recorded at the beginning, during the 3rd week, and at the end of each of the two periods. Exercise tests of the dogs were performed at the beginning and end of AC and questionnaire interviews with the owners were conducted at the end of each period. Study protocols were approved by the ethics committee of the Faculty of Veterinary Medicine, Chiang Mai University (S35/2561), and consent forms were obtained from the owners before beginning the experiment.

### 2.2. Physical Examination

A physical examination was performed to evaluate the health status of the dogs and to look for signs of abnormalities or illness.

### 2.3. Radiographic Study

Radiographs of the neck and thorax were taken at the right lateral and dorsoventral position. The tracheal size along the extrathoracic and intrathoracic airways were evaluated for any evidence of airway collapse.

### 2.4. HRV Determination

HRV was recorded with a Holter monitoring device (SEER™ 1000, GE Healthcare, Milwaukee, WI, USA) using standard chest leads for a 2 h period between 8 AM and noon. Dogs were kept in a resting cage and left undisturbed at ambient temperature of 25-degree Celsius. Time-domain and frequency-domain HRV were analyzed using the MARTM program (MAR program, GE Healthcare, Milwaukee, WI, USA).

### 2.5. Blood Collection

Three milliliters of venous blood was collected via venipuncture from either the cephalic or saphenous vein; then each sample was separated into three equal volume portions. The first portion was placed in tubes containing potassium ethylene diamine tetra-acetic acid (EDTA) for complete blood count analysis. The second portion was placed in tubes with lithium heparin for blood chemistry analysis, and the third portion was centrifuged at 3000 rpm for 15 min after which serum was collected and stored at −80 °C for malondialdehyde (MDA) analysis.

### 2.6. Oxidative Stress Determination

The HPLC-based assay was used to determine serum MDA levels in all-time courses. Serum MDA was measured at baseline and at week five. A 0.5 mL aliquot of serum was mixed with 1.1 mL of 10% trichloroacetic acid (TCA) containing BHT (50 ppm), heated at 90 °C for 30 min, and cooled down to room temperature. The mixture was centrifuged at 6000 rpm, 10 min. The supernatant (0.5 mL) was mixed with 0.44 M H_3_PO_4_ (1.5 mL) and 0.6% thiobarbituric acid (TBA) solution (1.0 mL) and then incubated at 90 °C for 30 min to generate a pink-colored product called thiobarbituric acid reactive substances (TBARS). The solution was filtered through a syringe filter (polysulfone type membrane, pore size 0.45 µm, Whatman International, Maidstone, UK) and analyzed with the HPLC system. The TBARS was fractionated on the adsorption column (Water Spherosorb ODS2 type, 250 × 4.3 mm, 5 µm), eluted with a mobile-phase solvent of 50 mM KH_2_PO_4_: methanol and detected at 532 nm. A standard curve was constructed from the peak from the height of standard 1,1,3,3-tetramethoxypropane (standard reagent for malondialdehyde) at different concentrations (0–10 µM). Plasma TBARS concentration was determined directly from the standard curve and reported as MDA equivalent concentration. MDA concentration in cardiac tissue and plasma were expressed in µM.

### 2.7. Acupuncture Procedure

Electroacupuncture (EA) was followed by aqua acupuncture (Aqua-AP) in this study. The acupuncture was performed by a certified veterinary acupuncturist (CVA) at the acupuncture clinic of the Small Animal Hospital, Chiang Mai University. Electroacupuncture involved stimulation of the GV-14, BL-12, BL-13, BL-23, ST-36, Baihui, Shen-shu and Fei-pan acupoints. Twenty-five mm 34-gauge sterile stainless-steel disposable acupuncture needles (zhong yan tai he) were used. Acupuncture needles were bilaterally connected to the electro puncturoscope (WQ-108, Dong Hua), except GV-14 and Baihui which were connected together. Selected acupoints were electrically stimulated with 2–4 Hz 3–4 mA 6 volt direct current for 15 min. After the first 15 min, the stimulation parameters were adjusted to 80–120 Hz of 3–4 mA 6 volt direct current for another 15 min. All of the dogs were physically restrained and comforted by a technician throughout the treatments.

Aqua-AC stimulation of the LU-1, CV-22, Fei-men, Fei-pan and Ding-chuan acupoints was performed after EA by injecting 0.5 mL of sterile 0.9% NaCl solution subcutaneously. The procedure was performed with 0.5-inch 30-gauge hypodermic needles.

### 2.8. Exercise Test

The exercise test was performed using a 6-min walk test (6MWT) protocol. Dogs were assigned to walk on a non-elevated flat treadmill at a speed of 1 km/h continuously for 6 min. During the exercise test, a Holter device was connected to the dogs and the information concerning HR was obtained from the record of continuous ECG data (MAR program, GE Healthcare, Milwaukee, WI, USA).

### 2.9. Quality of Life Determination Questionaire

The owners were asked to complete the questionnaires based on the clinical signs exhibited by their dogs. A questionnaire which provides a coughing score was used to evaluate quality of life at the end of each period. Owners were asked about the frequency and duration of coughing and their responses were scored. The maximum score from the questionnaire was eight points on a scale of 0 to 8. A high score indicates severe clinical signs, while a low score indicates less severe clinical signs. Coughing scores at the end of the NAC period and at the end of the AC period were then compared.

### 2.10. Statistical Analysis

HR during 6MWT data is presented in median and interquartile range. The others data are expressed as mean ± SE. For each group (NAC and AC groups), the differences in means of HRV parameters between the baseline (1st week), 3th week and 5th week was analyzed using ANOVA for repeated measurement data and the Tukey test was used for Post Hoc multiple comparisons. For AC group, the means of heart rate from the 6MWT were compared between baseline and the end of the 5th week using Two Sample paired *t*-test. Moreover, the difference in means of MDA levels for a particular group collected at baseline and the end of the 5th week were tested using One-way ANOVA. In addition, the quality of life data from the questionnaires were compared between the NAC and the AC groups at 5th week using ANOVA followed by Tukey test for Post Hoc tests. Assumptions of statistical analysis such as normality and homogeneity of variances were tested using the Shapiro-Wilk test and Levene’s test, respectively. The data were analyzed using program R Studio Software version 3.6.3. Statistical significance was set α = 0.05. 

## 3. Results

### 3.1. Clinical Baseline Characteristics of Dogs

Twenty TC dogs were enrolled in this study. The ratio of males to females was equal. Collapse of the trachea over the area of thoracic inlet was the most common finding (Table 1).

### 3.2. Effect of Acupuncture Treatment in the Function of the Autonomic Nervous System in Tracheal Collapse Dogs

The results of time-domain HRV analysis are shown in Figure 1. The mean NN at the 3rd week (645.98 ± 32.87) and 5th week (598.98 ± 31.01) of the AC treatment period were significantly increased when compared to the beginning of the AC treatment period (513.26 ± 35.10) (*p* < 0.05). The SDNN at the 3rd week (176.81 ± 14.90) and the 5th week (184.33 ± 15.80) of the AC treatment period were significantly increased when compared to the beginning of the AC treatment period (138.17 ± 13.46) (*p*< 0.05). The SDANN (45.86 ± 4.86) and ASDNN (127.95 ± 14.19) at the 5th week of the NAC period were significantly decreased when compared to the beginning of that period (62.35 ± 6.52 and 174.96 ± 18.70, respectively). The rMSSD at the 3rd week of the AC treatment period was significantly increased when compared to the beginning of the AC treatment period (66.70 ± 4.40 vs. 53.95 ± 4.48) (*p* < 0.05). However, the rMSSD at the 5th week of the NAC period was significantly decreased compared to the baseline (48.39 ± 3.90 vs. 62.43 ± 4.47) (*p* < 0.05). The pNN50 at the 3rd week (43.54 ± 3.49) and 5th week (40.40 ± 3.35) of the AC period were significantly increased compared to the beginning of AC period (30.98 ± 3.57) (*p* < 0.05).

The results of frequency-domain HRV analysis are shown in Figure 2. Total power at the 5th week of the NAC period was significantly decreased compared to the baseline (4876.50 ± 1203.73 vs. 7753.22 ± 1490.07) (*p* < 0.05). The LF/HF ratio at the 3rd week (1.06 ± 0.14) and the 5th week (0.92 ± 0.13) of the AC period were significantly decreased compared to the beginning of the AC period (1.37 ± 0.14) (*p* < 0.05). Moreover, the acupuncture treatment significantly increased HF and VLF compared to the baseline. This study demonstrated that AC increased mean NN, SDNN, rMSSD, and pNN50 in the time-domain HRV, while it decreased the LF/HF ratio of the frequency-domain HRV of tracheal collapse dogs.

### 3.3. Effect of Acupuncture Treatment on the Oxidative Stress (Serum MDA Level) in Tracheal Collapse Dogs

The mean serum concentration level of MDA was not significantly altered during the NAC period but decreased significantly during the AC period (Table 2).

### 3.4. Acupuncture Improved Exercise Performance by Lowering HR during 6MWT in TC Dogs

The mean HR before initiation of AC treatment was 184 ± 6.98 bpm. The mean HR at the end of the AC period (5th week) was 135 ± 6.55 bpm which was significantly lower than at the beginning of the AC period (*p* < 0.05) (Figure 3).

### 3.5. Effect of Acupuncture Treatment on the Quality of Life as Measured by Reduced Coughing Scores

The quality of life scores from the questionnaire at the end of the NAC and the AC periods were 6.35 ± 0.27 and 1.45 ± 0.37, respectively. The coughing score of the AC period was significantly lower than in the NAC period (*p* < 0.05). This suggests that AC can reduce coughing scores in TC dogs.

## 4. Discussion

The present study demonstrated the positive effects of acupuncture on TC dogs at the cellular level (reduced MDA level), functional level (improved cardiac autonomic balance), and clinical level (reduced coughing and, consequently, improved quality of life).

Acupuncture might improve HRV by decreasing neuropeptide expression in the paraventricular nucleus of the hypothalamus leading to a reduction of the sympathetic outflow as was demonstrated for rats [45]. A previous study on dogs reported that acupuncture at ST-36 produced a parasympathomimetic-like effect and increased the HF parameter. The raising of HF value reflects the improvement of the vagosympathetic activity with the parasympathetic system predominant [46]. Based on the improvement of HRV in this study, we suggest that acupuncture treatment can reduce cardiac autonomic dysfunction by modulating the balance between the sympathetic and parasympathetic nervous systems [47,48] in tracheal collapse dogs. The positive effect of acupuncture has also been reported in the treatment of several abnormalities such as cardiac arrhythmia and hypertension, asthma, allergic rhinitis, allergic bronchial, and chronic obstructive pulmonary disease (COPD) in both humans and various animals [49,50,51,52]. The sympathetic outflow from the central nervous system is partly regulated by the gracile nucleus and mNTS in the hypothalamus. Acupuncture has been shown to increase the expression of neuronal nitric oxide synthase (nNOS) in these nuclei, leading to a reduction of sympathetic outflow [53,54]. Excessive ROS also reduces NO production. An increase of NO results in a decrease in central and peripheral sympathetic activity [55]. A previous study on rats demonstrated that a reduction of ROS resulting from AC therapy decreased the level of MDA in brain tissue [56]. Acupuncture may also decrease ROS production in peripheral tissues and increase nNOS production. The mechanism by which AC attenuated HRV impairment in the present study may have been the decrease in sympathetic activity resulting from an increase in the production of nNOS.

A previous study on obese male dogs demonstrated that a high level of plasma MDA impaired HRV, while a low level of plasma MDA improved HRV [57]. A decrease in MDA level may indirectly indicate a dominance of the parasympathetic nervous system [58]. The reduction of MDA level in TC dogs additionally treated with acupuncture reported in this study is in concordance with the result of previous studies which demonstrated that acupuncture could reduce oxidative stress in an animal model with ischemic heart disease and obesity [54,59]. Oxidative stress also increases during the inflammatory process [60,61]. Coughing resulting from tracheal collapse irritates the airway and leads to inflammation of the trachea and pulmonary tissues [62,63]. Previous research has demonstrated that acupuncture treatment reduces oxidative stress damage of mitochondrial membrane lipids via elevated activity of endogenous antioxidants and decreased level of MDA which prevents the lipid peroxidation process [64]. The cough reflex can be induced by stimulation of the trachea [65]. Stimulation of this reflex is mediated by activation of C-fibers in the airway [66]. According to the gate control theory of Melzack and Wall, sensation originating from AC may activate inhibitory interneurons in the spinal cord and inhibit sensory impulses conducted by C-fibers [67]. AC treatment in TC dogs may produce the same effect on the C-fibers in pulmonary tissues. Inhibition of sensory input from C-fibers in the airway in TC dogs might be a mechanism by which AC alleviates coughing in TC dogs.

A previous study reported that HR after a 6-min walk test (6MWT) in dogs with myxomatous mitral valve disease was higher than that of the normal healthy dogs [68]. Pongkan et al., 2020 reported that the HR in obese dogs was higher than in normal BCS animals [61]. The dogs used in our study were obese and presumably had higher heart rates than normal weight dogs [61]. The reduction of HR after AC in this study may have resulted from a shift in ANS balance toward parasympathetic predominance. Suppression of the cardiac sympathetic nervous system is coordinated with activation of cardiac vagal activity which reduces the heart rate [51]. Reduction of HR in a 6MWT in TC dogs may be used to monitor improvement during treatment in routine clinical practice. Since the lack of a control group in this study, we cannot ultimately discard a possibility that the state of the dogs improved simply as a function of time from the start of the experiment. Further study with an additional control group might be warranted to prove the validity of the acupuncture.

## 5. Conclusions

Our study showed that application of acupuncture treatment improved HRV and quality of life, while reducing serum MDA levels and HR after exercise tests in TC dogs. The positive therapeutic effects of AC found in this study might be caused by several mechanisms including modulation of ANS activity leading to predominance of the parasympathetic activity and reduction of systemic oxidative stress. The study suggests that AC treatment might be a suitable adjunct to treatment for TC disease.

## Figures and Tables

**Figure 1 vetsci-09-00088-f001:**
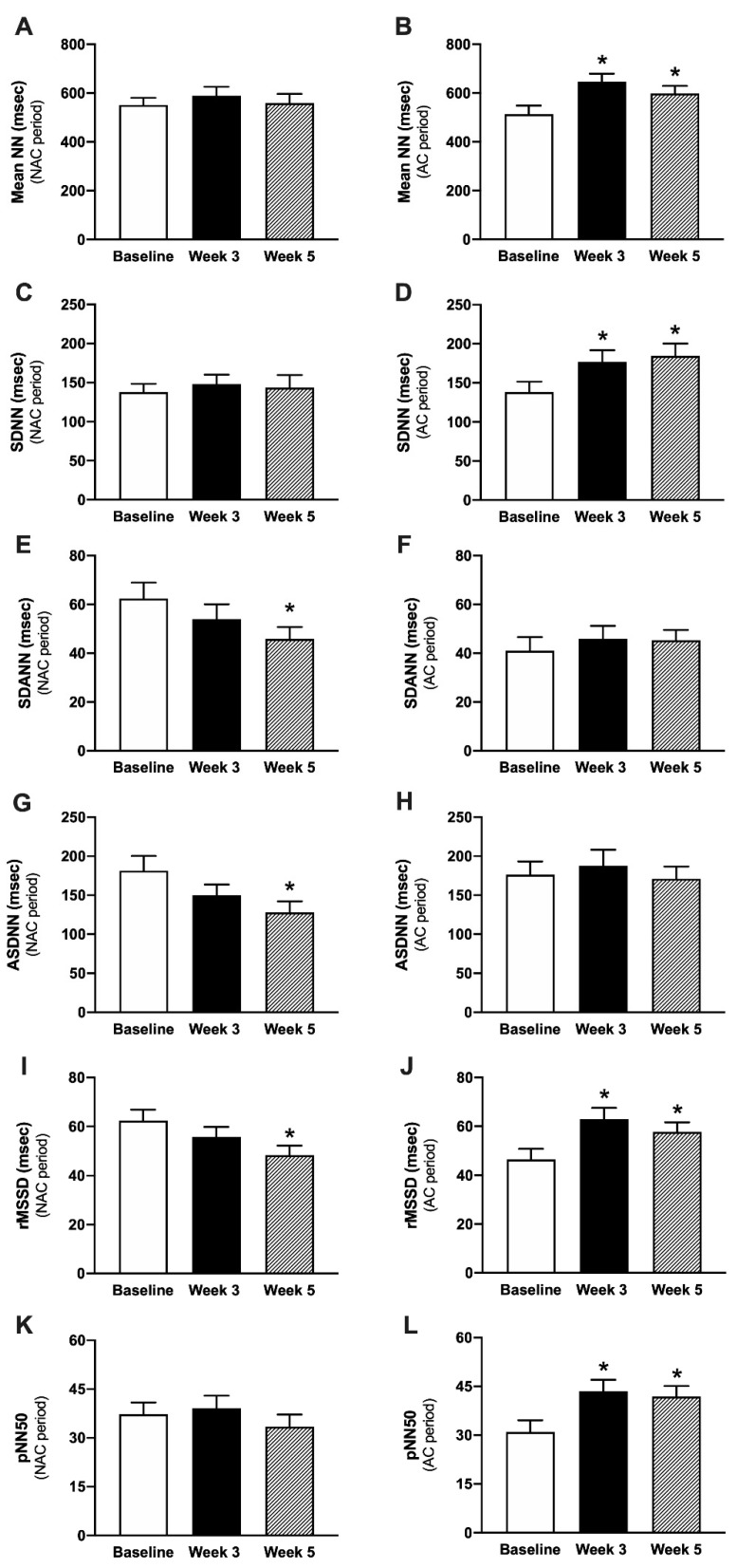
Time-domain parameters of HRV in the NAC (**A**,**C**,**E**,**G**,**I**,**K**) and AC (**B**,**D**,**F**,**H**,**J**,**L**) period. Abbreviations: NAC period = no acupuncture treatment period, AC period = acupuncture treatment period, mean NN = average NN intervals, SDNN = standard deviation of all NN intervals, SDANN = standard deviation of the averages of NN intervals in 5-min, ASDNN = average standard deviation of all 5-min R-R intervals, rMSSD = root mean square of the sum of the squares of differences between adjacent NN intervals, pNN50 = percentage of successive NN intervals > 50 msec, * *p* < 0.05 comparison with the baseline.

**Figure 2 vetsci-09-00088-f002:**
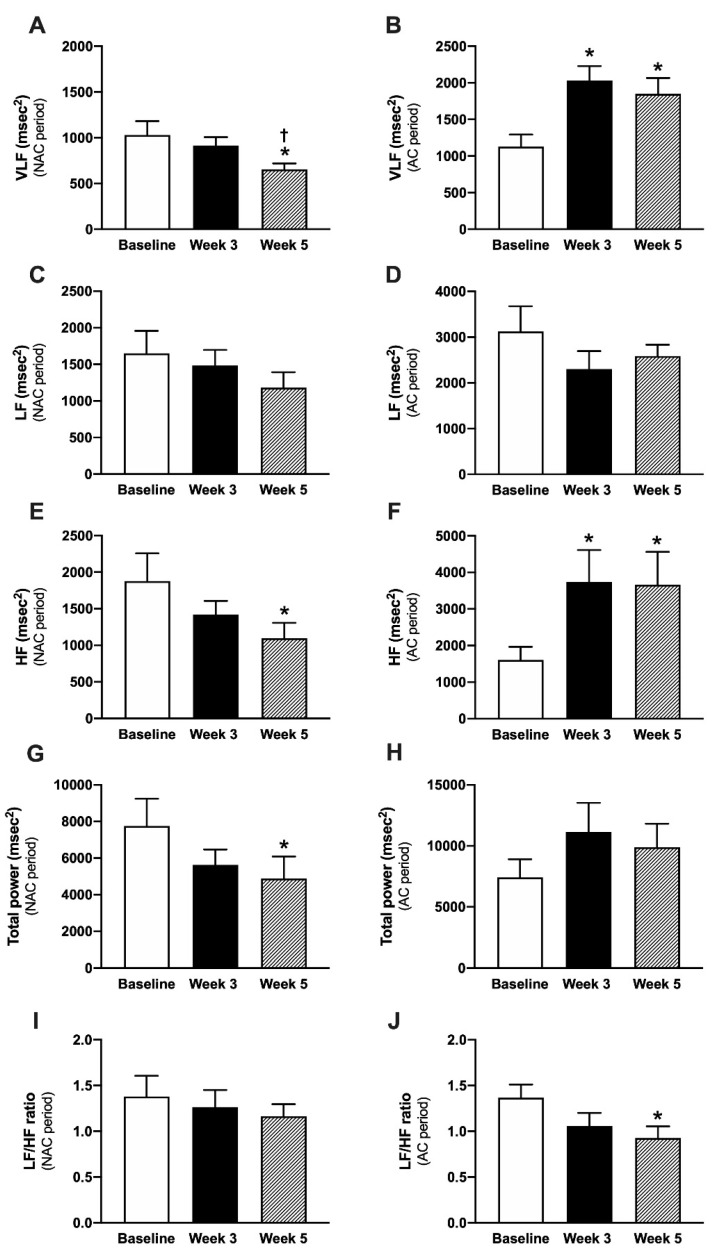
Frequency domain parameters in the NAC (**A**,**C**,**E**,**G**,**I**) and the AC (**B**,**D**,**F**,**H**,**J**) periods. Abbreviations: NAC period = no acupuncture treatment period, AC period = acupuncture treatment period, VLF = very low frequency, LF = low frequency, HF = high frequency, TP = total power, * *p* < 0.05 comparison with the baseline, † *p* < 0.05 comparison with the 3rd week.

**Figure 3 vetsci-09-00088-f003:**
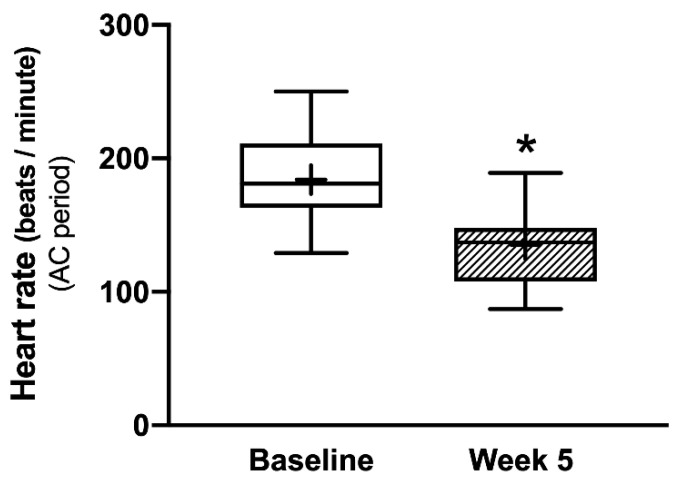
A box plot of heart rate (HR) during 6MWT in TC dogs. * *p* < 0.05 comparison with the baseline.

**Table 1 vetsci-09-00088-t001:** Clinical baseline characteristics of the tracheal collapse dogs at the beginning of the study (*n* = 20).

Parameter	Mean (SE)	Range
Age (years)	6.8 (0.90)	2–16
Body weight (kg)	5.18 (0.69)	1.8–11.8
Body condition score	7.4 (0.13)	7–9
Sex		
Female (not spayed) (No., %)	5, 25	
Female (spayed) (No., %)	5, 25	
Male (not castrated) (No., %)	4, 20	
Male (castrated) (No., %)	6, 30	

**Table 2 vetsci-09-00088-t002:** Serum MDA levels in tracheal collapse dogs (*n* = 20) at the beginning and the end of NAC and AC period.

*n* = 20	NAC Period		AC Period	
	At Baseline	End of 5th Week	At Baseline	End of 5th Week
MDA level (µM/L)	1.68 ± 0.18	1.62 ± 0.12	1.78 ± 0.17	1.36 ± 0.12 *

Abbreviations: MDA = malondialdehyde, µM/L = micromoles per liter, NAC period = no acupuncture treatment period, AC period = acupuncture treatment period, * *p* < 0.05 comparison with the beginning of the AC period.

## Data Availability

The data presented in this study are available on request from the corresponding author.

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
