# Peer review of "Acupuncture Improves Heart Rate Variability, Oxidative Stress Level, Exercise Tolerance, and Quality of Life in Tracheal Collapse Dogs"

_vetsci, 2022, doi:10.3390/vetsci9020088_

Round 1
Reviewer 1 Report
This seems to be a well performed and presented study. Although interest to the readers of European and North American countries may not be high currently, I do believe that this will soon change... Europe at least seems to be willing to learn from the Asian countries once again. I have a number of rather minor comments wich I hope will help the authors to improve the quality of their manuscript; all are included in the attached file

Author Response
Dear reviwer1
Thank you very much for the valuable suggestion to strengthen our manuscript. We try to correct all the reviewer suggestions and was highlighted the change in yellow color. Please see the attachment for the revised version.
Sincerely yours,
Dr.Chavalit Boonyapakorn
Corresponding author

Reviewer 2 Report
This is an interesting and well written paper and its conclusions are important for veterinary practice and the welfare of small dogs suffering from tracheal collapse disease. However, some aspects of this paper may be seriously improved.
Important general remark: in this study all dogs were subjected first to NAC conditions (no acupuncture treatment), and then to AC conditions (acupuncture treatment). Such experimental design lacks a control group: dogs that continued to be subjected only to NAC conditions during the whole experiment. Without such a control group, we cannot ultimately discard a possibility that the state of the dogs improved simply as a function of time from the start of the experiment, and not as a consequence of the received acupuncture treatment. I agree that in light of the obtained results such an eventuality seems to be little probable, but the authors should honestly mention it in the discussion of their results, and point out that without such a final proof the validity of the conclusions of the experiments reported in the present study must be considered an open question, to be solved in future research.
All dogs were treated with two acupuncture techniques, electroacupuncture and then aqua acupuncture. In a future study it might be interesting to compare the effects of these two types of acupuncture.
The authors should provide the information about dog breeds in the chapter devoted to Methods, and not in the description of the Results. They also mentioned in the Discussion that the studied dogs were obese. This information also should have been provided in the Methods.
It would be helpful for the readers if the authors could name the principal elements of the physical examination of the dogs, and explain more precisely what clinical symptoms corresponded to successive „quality of life scores”, and what is the exact relationship between the notions "quality of life scores" and "coughing scores".
Two sentences at the start of the Results (lines 170-172) are a part of the instructions for the authors, and not of the text of their paper. They should be deleted.
The quality of Figures 1 and 2 is very low. It is almost impossible to decipher what is written next to the axes Y.
The same data are presented in Figures 1 and 2 and reported in great detail in the text of the paper, together with conclusions concerning presence/absence and directions of changes. These conclusions are also repeated in the captions of Figures 1 and 2. It is usually recommended to avoid such redundance of information in scientific papers.
So far, the description of methods contains only the information that "data are expressed as mean ± SE." However, the data shown in Figure 3 are medians, quartiles and range.
An important question worth to be discussed: did all the tested dogs or only a part of them respond to acupuncture treatment? If the tested sample included also non-responders (how many?), it might be useful to have a closer look on these individuals, and to try to find out what might have decided that they did not respond to AC treatment.
I could not see supplementary materials - the link provided in the manuscript did not work
The appendix with the explanations of the abbreviations used in the text is useful. However, it is not complete. Here are other abbreviations that should have been included, too: AC, ANS, BCS, COPD, NAC, NLF, NN, nNOS, R-R. This appendix should have been announced earlier in the text.
I also made many other comments (in total 145). All my comments can be found in pop-up notes attached directly to the attached manuscript of the reviewed paper. Some of them are related to the linguistic side of the reviewed paper, which is rather good but not perfect.

Author Response
Dear reviewer2
We would like to thank the reviewer for all of the helpful advice to strengthen the manuscript. We responded point by point to the reviewer's suggestion as shown in the table below. The change in the revised manuscript as the reviewer suggestion was highlighted in light blue color. Please see the revised manuscript in the attachment file
Sincerely yours,
Dr.Chavalit Boonyapakorn
Corresponding author.
Reviewer’s suggestion |
Response |
This is an interesting and well-writte paper and its conclusions are important for veterinary practice and the welfare of small dogs suffering from tracheal collapse disease. However, some aspects of this paper may be seriously improved. |
- |
Important general remark: in this study all dogs were subjected first to NAC conditions (no acupuncture treatment), and then to AC conditions (acupuncture treatment). Such experimental design lacks a control group: dogs that continued to be subjected only to NAC conditions during the whole experiment. Without such a control group, we cannot ultimately discard a possibility that the state of the dogs improved simply as a function of time from the start of the experiment, and not as a consequence of the received acupuncture treatment. I agree that in light of the obtained results such an eventuality seems to be little probable, but the authors should honestly mention it in the discussion of their results, and point out that without such a final proof the validity of the conclusions of the experiments reported in the present study must be considered an open question, to be solved in future research. |
The discussion section had been revised according to the concerning point (lines305-309). |
All dogs were treated with two acupuncture techniques, electroacupuncture and then aqua acupuncture. In a future study it might be interesting to compare the effects of these two types of acupuncture. |
- We agree with the suggestion. |
The authors should provide the information about dog breeds in the chapter devoted to Methods, and not in the description of the Results. They also mentioned in the Discussion that the studied dogs were obese. This information also should have been provided in the Methods. |
Corrected (lines81-83) |
-It would be helpful for the readers if the authors could name the principal elements of the physical examination of the dogs,
- and explain more precisely what clinical symptoms corresponded to successive „quality of life scores”, and what is the exact relationship between the notions "quality of life scores" and "coughing scores". |
- Corrected (lines106-107)
|
Two sentences at the start of the Results (lines 170-172) are a part of the instructions for the authors, and not of the text of their paper. They should be deleted. |
corrected |
The quality of Figures 1 and 2 is very low. It is almost impossible to decipher what is written next to the axes Y. |
Thank you for the suggestion. The new figures were attached as extra files. Now, all of the figures met the standard requirement for publication. |
The same data are presented in Figures 1 and 2 and reported in great detail in the text of the paper, together with conclusions concerning presence/absence and directions of changes. These conclusions are also repeated in the captions of Figures 1 and 2. It is usually recommended to avoid such redundance of information in scientific papers. |
We would like to thank the reviewer for their valuable comment. We minimized the redundancy of information by deleting the repeated wording in the caption of figure 1 and figure2. |
So far, the description of methods contains only the information that "data are expressed as mean ± SE." However, the data shown in Figure 3 are medians, quartiles and range. |
The statistical analysis section had been revised for more clarity as suggested by the reviewer. (lines 164-165) |
An important question worth to be discussed: did all the tested dogs or only a part of them respond to acupuncture treatment? If the tested sample included also non-responders (how many?), it might be useful to have a closer look on these individuals, and to try to find out what might have decided that they did not respond to AC treatment. |
Fortunately, all of the dogs responded well to the acupuncture treatment. According to our small sample size, further study with a larger population might show the effect of acupuncture more clearly. |
I could not see supplementary materials - the link provided in the manuscript did not work |
Corrected (no supplementary materials) |
The appendix with the explanations of the abbreviations used in the text is useful. However, it is not complete. Here are other abbreviations that should have been included, too: AC, ANS, BCS, COPD, NAC, NLF, NN, nNOS, R-R. This appendix should have been announced earlier in the text. |
Corrected. |
I also made many other comments (in total 145). All my comments can be found in pop-up notes attached directly to the attached manuscript of the reviewed paper. Some of them are related to the linguistic side of the reviewed paper, which is rather good but not perfect. |
We would like to thank the reviewer again for spend time to strengthen our manuscript. Pop-up notes were used in the revised version as label with the light-blue color |

Reviewer 3 Report
General comment: The authors presented an interesting review work concerning the role of acupuncture in dogs with tracheal collapse.The manuscript is written in a comprehensive way and it is well structured.
Title: The title is concise and adequate.
Abstract: It is adequate. Please provide the full name before the abbreviation “MDA”. The keywords should be different from those used in the title.
Introduction: It is adequate. The authors provided an adequate overview of the thematic.
Once defined, the abbreviations should be used throughout the manuscript.
Methods: They are adequately described.
Results: The quality of Figures 1, 2 and 3 should be improved.
All values presented in the Results should present dot and not comma (e.g. page 5, lines 197-198).
Methods: The methods are well described.
Discussion and Conclusion: The discussion is adequate. The conclusion is based in the results of the studies included in the review.
Recommendation: The manuscript should be accepted for publication after a minor revision.
Author Response
Dear reviewer3
Thank you very much for your valuable suggestions. The change in the revised version as your suggestion was highlighted in green color. Please see the attachment for the revised version.
Sincerely yours,
Dr.Chavalit Boonyapakorn
Corresponding author
